# Strategies to Improve Nanofibrous Scaffolds for Vascular Tissue Engineering

**DOI:** 10.3390/nano10050887

**Published:** 2020-05-05

**Authors:** Tianyu Yao, Matthew B. Baker, Lorenzo Moroni

**Affiliations:** Complex Tissue Regeneration Department, MERLN Institute for Technology Inspired Regenerative Medicine, Universiteitssingel 40, 6229ER Maastricht, The Netherlands; t.yao@maastrichtuniversity.nl (T.Y.); m.baker@maastrichtuniversity.nl (M.B.B.)

**Keywords:** nanofibers, scaffolds, vascularization

## Abstract

The biofabrication of biomimetic scaffolds for tissue engineering applications is a field in continuous expansion. Of particular interest, nanofibrous scaffolds can mimic the mechanical and structural properties (e.g., collagen fibers) of the natural extracellular matrix (ECM) and have shown high potential in tissue engineering and regenerative medicine. This review presents a general overview on nanofiber fabrication, with a specific focus on the design and application of electrospun nanofibrous scaffolds for vascular regeneration. The main nanofiber fabrication approaches, including self-assembly, thermally induced phase separation, and electrospinning are described. We also address nanofibrous scaffold design, including nanofiber structuring and surface functionalization, to improve scaffolds’ properties. Scaffolds for vascular regeneration with enhanced functional properties, given by providing cells with structural or bioactive cues, are discussed. Finally, current in vivo evaluation strategies of these nanofibrous scaffolds are introduced as the final step, before their potential application in clinical vascular tissue engineering can be further assessed.

## 1. Introduction

Tissue engineering, which combines the principles and methods of the life sciences with those of engineering to create tissue replacements to direct tissue regeneration, has attracted many researchers with the hope of regenerating a patient’s own tissues and organs without the need for tissue/organ transplantation [1,2]. The most classical tissue engineering approach consists of cell seeding on a scaffold, followed by cell proliferation, differentiation (if the starting cells are stem cells), and tissue formation through extracellular matrix synthesis. The resulting biological construct is typically matured in vitro to become a functional new tissue that can be implanted back in the host environment [3]. 

The field of regenerative medicine has achieved significant progress in the past decade (Figure 1) [4]. A few examples include: (1) the generation of induced pluripotent stem (iPS) cells from adult somatic cells, which showed the possibility for personalized regenerative medicine [5,6,7]; (2) the development of scaffolds with tailored stiffness and topography that could regulate stem cell differentiation, providing an approach to control cell phenotype using physical and mechanical cues [8,9,10]; (3) a more fundamental understanding of the role of immune cells in presence of a regenerative medicine treatment (e.g., at the interface with biomaterials, or triggered by the application of cell therapy) [11,12,13]. Some successful products to aid in tissue regeneration are already available in the clinic, such as poly(vinyl alcohol) sheets developed for vessel coverage during anterior vertebral surgery [14], collagen sponges with β-tricalcium phosphate (β-TCP) commonly used as void fillers for bone regeneration [15], and the use of limbal stem cells for corneal injury repair [16].

However, significant challenges remain before the widespread adoption of tissue engineering approaches in the clinic. Several important aspects should be considered for successful regeneration, such as (i) scaffold design with desired mechanical, chemical, and biological properties that better mimic a tissue’s native microenvironment and better support cell activity; (ii) maintenance and regulation of growth factors to steer cellular behavior; and (iii) vascularization, allowing integration of the engineered tissues into the host system for fully functional regeneration [17,18]. 

Tissue engineering approaches typically start with scaffold design. An engineered scaffold should provide favorable biochemical (e.g., surface chemistry [19]) and biophysical cues (e.g., fibrous structure [20], hydrophilicity [21], and stiffness [22]) to mimic the native extracellular matrix (ECM) for cells. Biochemical and biophysical properties from ECM are important to support cell growth and affect cell functions [23]. For example, scaffolds mimicking key features of the ECM can regulate cell behavior, including attachment, migration, proliferation, and differentiation in tissue regeneration [24,25,26]. The creation of scaffolds that can better mimic the native ECM and provide native structures has become a common goal for tissue engineering [27]. In native tissues, the diameters of structural ECM proteins are smaller than those of the cells, approximately 50–500 nm [28]. ECM normally provides structural support and biological factors to guide cell maturation and integration to form tissues [29,30,31]. To mimic the nanofibrous structure of ECM, three fabrication techniques have been mostly investigated: molecular self-assembly [32,33,34,35], phase separation [36,37], and electrospinning [38,39,40]. These three methods are briefly introduced in the following section and the comparisons between methods are listed in Table 1.

## 2. Nanofibrous Fabrication Techniques

### 2.1. Self-Assembly

Self-assembly is a method by which simple molecular and macromolecular building blocks are engineered to organize themselves through non-covalent forces, such as hydrogen bonding, hydrophobic forces, and electrostatic interactions [41,42]. The molecules and the intermolecular forces are critical for the self-assembly of nanofibous scaffolds. Self-assembly of biomolecules to form nanostructures is widely used in natural systems, as can be seen in many of the cells or ECM superstructures [43]. Likewise, synthetically created self-assembling systems and materials are widely used in tissue engineering for a variety of functions [44,45,46]. A well-established approach to creating molecularly self-assembled biomaterials is to engineer amphiphilic peptide sequences [47,48,49]. The self-assembly of these peptide amphiphiles is controlled by both hydrophobic interactions and hydrogen bonding [50]. These peptide amphiphiles (PA) each consist of a hydrophobic tail group and a hydrophilic head group (Figure 2) [51,52]. During the self-assembly process, the hydrophobic tails of PA will be driven away from water and towards each other, and smaller nanofibers around 5–25 nm can be successfully achieved based on the peptide design [53]. Peptide amphiphiles remain one of the most prolific platforms for molecularly self-assembled fibrous scaffolds and have found success in applications ranging from spinal cord regeneration to vascularization [54,55]. 

Supramolecular nanofibers can also be created by other self-assembling motifs. An early example of supramolecular polymers forming nanofibers is ureido-pyrimidone (Upy). Sijbesma and coworkers prepared the self-assembled UPy nanofibers through self-complementary four-hydrogen bonding between self-complimentary Upy moieties [56]. The fibrous Upy based materials have been widely used for drug delivery and tissue engineering [57,58], with early success in regenerative heart valves [58]. The self-assembly process is based on non-covalent interactions, which show the reversible nature and can create responsive materials. Advantages for Upy materials include their ability to adapt their structure to the activity of cells, provide enhanced mechanical properties via the H-bonding motif, and modulate co-polymerization via self-assembly. However, the delicate control of non-covalent interactions needing to be sufficiently strong, but still reversible, is still challenging, and rational design is in its infancy. Other examples of nanofibrous self-assembling materials include bsenzene-1,3,5-tricarboxamide (BTA) [59,60], polyelectrolytes [61], and squaramide derivatives [62]; these supramolecular motifs are not well explored in tissue engineering, yet hold promise as alternative platforms. 

The nanofibers produced by molecular self-assembly usually are much smaller than those produced by electrospinning, clearly due to the molecular design character of self-assembly [32]. While fascinating helical nanofibers can be particularly well-formed by the self-assembly method, full and rational control of the self-assembling of peptides or supramolecular polymers at the nanoscale is still a challenge. For example, it is hard to control nanofibers assembling in a direct way (e.g., guiding nanofiber deposition like electrospinning). Another challenge is the scaling of self-assembled nanofibers in comparison to polymer processing techniques. A study to investigate how to control and produce self-assembled scaffolds at large scales will be beneficial for the further application of this technique in tissue engineering [63].

### 2.2. Thermally Induced Phase Separation

Thermally induced phase separation (TIPS) has been used to produce three-dimensional (3D) porous scaffolds since the 1980s. Nanofibrous materials have also been made by this technique [37]. When phase separation occurs, two different phases, a polymer-rich phase and a polymer-poor phase, are divided in polymer mixtures. The polymer-rich phase dries during solvent evaporation and then forms 3D porous structures at a certain concentration. If the polymer concentration is too low, polymer beads will be formed after phase separation. High concentration will only lead to forming small pores without interconnection in polymer scaffolds. Hence, the polymer concentration for a specific polymer–solvent system, with which TIPS is performed, is an important parameter to be optimized. TIPS involves four main steps: (a) dissolution of a polymer in a targeted solvent to form a homogeneous polymer solution, (b) separation of two phases, (c) solvent evaporation and polymer solidification, and (d) freeze-drying of the final product (Figure 3) [65,66]. 

One of the advantages of TIPS is that adjusting temperature and polymer concentration could change the pore sizes and fiber diameters of scaffolds [67]. Ma and his team have fabricated poly (*L*-lactic acid) (PLLA) nanofibrous matrices by using TIPS with a low polymer concentration from 1% to 7.5% in tetrahydrofuran [37]. The diameters of PLLA nanofibers were 50–500 nm and the matrix porosity reached 98.5%. Normally, using low-concentration polymer solutions could produce smaller fibers and highly porous scaffolds. The porous nanofiber scaffolds formed by phase separation, which were reported to mimic the structure of collagen type I fibers in ECM, are useful for tissue engineering applications [68]. For example, Yang et al. fabricated PLLA fibrous scaffolds via phase separation for nerve regeneration [69]. The nano-fibrous scaffold with an average fiber diameter of 200 nm was developed to better mimic the natural ECM environment. In vitro results showed that PLLA nanofibrous scaffolds could support neural stem cell (NSC) growth and differentiation. A single neuron from its cell body to tip of the neurite was clearly observed on the fibrous scaffolds. Hence, authors concluded that the nanometric features of the scaffold could provide cells physical cues, which favored stem cell differentiation and neurite outgrowth. However, the quantification of neuron length on different scaffolds and the effect of fiber diameter on neuron growth remains to be investigated. 

There are still limitations in TIPS. Firstly, the fibers created by TIPS are randomly oriented and the diameter range is wide. Secondly, phase separation to produce fibrous scaffolds is limited to the laboratory scale. Alternative methods to control and upscale TIPS fibrous scaffolds are needed for tissue engineering to bring this technology towards more clinical adoption. 

### 2.3. Electrospinning

Electrospinning is a common and cost-effective fabrication technique to generate nano/micro fiber scaffolds [70]. The fiber formation process is driven by the application of an electrostatic force on a polymer solution [38]. When electrostatic forces are higher than the surface tension of the polymer solution, a polymer droplet is stretched to form the so-called Taylor’s cone, as a charged jet (Figure 4). After the charged polymer solution is sprayed out, the solvent evaporates quickly and polymer fibers solidify. Then, the elongated fibers are deposited to a grounded collector. 

A broad variety of natural and synthetic polymers have been reported to produce nanofibers via electrospinning. Due to the wide applicability of electrospinning to a variety of biomaterials, this technique has gained a lot of attention in tissue regeneration [71]. The major advantage of electrospun scaffolds is the high porosity and small structural features mimicking the physical dimensions of ECM [72]. By controlling the processing parameters of electrospinning (including polymer content, high voltage, flow rate, and collecting distance), the structure, diameter, and pore size of nanofibers can be successfully controlled [38,73]. 

Many researchers have highlighted that the micro and nanofibrous structures in electrospun scaffolds imitate the morphological features of the ECM, and offer a viable environment to help cells attach to, proliferate on, and migrate on [74,75]. Therefore, a considerable number of electrospun scaffolds have been studied for tissue engineering applications, including some for the regeneration of soft tissues, such as that of skin [76], blood vessels [77], cartilage [78], and nerves [79], and hard tissue regeneration, for bone [80,81,82].

## 3. Nanofiber Structure Design

Although there are three main approaches used for nanofiber fabrication, as mentioned above, electrospinning is the most common and effective method for generating nanofibers for tissue engineering applications. Traditional electrospun nanofibrous scaffolds are randomly oriented due to the jet instability during electrospinning. In past years, researchers have attempted to make nanofibrous scaffolds with highly ordered structures by changing the electrospinning processing parameters or using additional setups. Such variations have also resulted, however, in mixtures of fibers and beads. Although a beaded structure in electrospun scaffolds has often been considered as a defect structure of the produced fibers, some researchers proposed the use of a uniform beaded nanofibrous scaffold structure for drug delivery applications [83]. Examples comprise using lower concentration (5%) of poly(ethylene oxide) (PEO) solution, which could result in a more beaded structure compared with higher PEO concentrations [84]. Highly aligned PLLA fibers have been produced by using a rotating disk as a collector [85]. The tunability of nanofiber structures provides more possibilities for the application of electrospun scaffolds in tissue engineering applications. In this section, some of the most relevant achievements in fiber architectures, including fabricating beaded, aligned, honeycomb, web and wavy fibers, and nanotubes, will be described. 

### 3.1. Porous Microsphere/Nanofiber Composites

It is well known that the lotus leaves are dry and exhibit self-cleaning effects from the mud. The hierarchical micro/nanostructure of the lotus-leaf leads to a specific surface roughness, which is highly related to its super hydrophobicity [86]. Inspired by the lotus-leaf, many researchers have been investigating its surface structure and attempted to mimic its super hydrophobic surface for biomedical applications. Super hydrophobic surfaces were reported to restrict protein adsorption and cell adhesion [87]. Some researchers showed the potential of creating a super hydrophobic micropattern or a tunable hydrophobic surface to control cell behavior [88,89]. Moreover, super hydrophobic materials could be applied in drug delivery, to slow down drug release. Kaplan and his colleagues fabricated polycaprolactone (PCL) and poly(glycerol monostearate-co-caprolactone) (PGC) electrospun nanofiber meshes, which showed super hydrophobic surface properties [90]. The non-wetting properties made them successful as a drug delivery system to sustain the release of cisplatin over 90 days. Longer-term studies and further biocompatibility of these platforms are needed to mature this approach. 

Electrospinning is an effective method with which to prepare super hydrophobic surfaces by fabricating a composite hierarchical structure, which contains microspheres and micro/nanofibers inside. Jiang et al. created electrospun polystyrene (PS) porous microspheres and nanofiber composites in order to obtain super hydrophobicity. The contact angle of fabricated PS films was higher than 150°, confirming super hydrophobic properties (Figure 5A) [91]. By adjusting the concentration of the starting solution, the morphologies of PS films could be controlled. However, no cell study was reported on these superhydrophobic PS films. Understanding the exact role of superhydrophobic surfaces to control cell behavior is highly needed for the potential application of superhydrophobic films in tissue engineering.

The shapes and morphologies of beads and fibers can be controlled by adjusting electrospinning parameters. For example, Lee et al. illustrated that changing polymer concentration during electrospinning could result in different bead dimensions and morphology. Bead structures were largely produced when electrospinning with a PS solution below 15%, while only fibers were formed after increasing the concentration above 15% [92]. As PS is not biodegradable, translating these findings to biodegradable polymers for a hybrid fiber-beaded scaffold fabrication could find better use for tissue engineering. Hence, electrospinning polymer solutions with lower concentrations to produce beads or beaded fibers was also verified with several other polymers [93,94,95]. Micro or nanostructured electrospun meshes are known to be more hydrophobic, which is mainly ascribed to the underlying beads or fibrous structures producing air pockets between the mesh surface and water droplets when measuring the contact angle [96,97,98]. Most of the cell studies in tissue engineering have been performed on uniformly electrospun micro/nano fibers; the effects of electrospun meshes containing beads remain relatively unexplored. 

### 3.2. Aligned Nanofibers

Aligned nanofibers provide specific physical cues able to directly influence cell activity, especially by guiding cell distribution and elongation in the direction of the fibers [99]. Several aligned electrospun fibers have been used in tissue regeneration. Deepthi et al. fabricated electrospun aligned PLLA nanofibers and covered them with a chitosan-collagen hydrogel in order to mimic the fibrous ECM for tendon regeneration [100]. In the bone matrix, collagen fibers are preferentially oriented (parallel to the long axis of bone), and their structural features are important to provide strength in tension and resistance during bone bending [101,102,103]. Many researchers attempted to mimic this unique structure of bone by using aligned electrospun fibers [104,105]. The effects of fiber alignment on osteogenic differentiation have been widely investigated [82,106,107]. The alignment of fibers was reported to induce osteogenic differentiation and upregulate the gene expression of osteogenic markers, including osteocalcin, runx2, osteopontin, and osteonectin, compared to random fibers [104]. Aligned electrospun fibers provide specific geometrical cues, which guide the structure and function of human mesenchymal stromal cells (hMSCs) for bone regeneration [104,108]. Calcium content on aligned fibers was significantly higher compared to that of random fibers [82]. 

Moreover, aligned fibers also showed a positive effect on vascular regeneration. Zhu et al. fabricated PCL aligned fibers by using a rotating mandrel to collect fibers [109]. Human umbilical artery smooth muscle cells (huSMC) and human umbilical vein endothelial cells (HUVECs) were seeded on aligned PCL fibers separately, showing that the alignment of fibers guided cell distribution and elongation, which is important for tube formation. The elongated cells could be due to cell pseudopodia stretched along the fiber direction. However, indications showing that the aligned topography effectively guided tube formation of endothelial cells remained elusive. The co-culture of hMSCs and HUVECs on aligned fibers could be added in this study to further investigate the potential influence of hMSCs in HUVECs tube formation on aligned fiber. In a different study, Wu et al. produced 3D PCL fibrous tubular scaffolds [110] and proved that fiber alignment effectively restricted the endothelial cells’ distribution. The alignment of fibers could be, therefore, used to create functional vascular grafts to achieve mechanical compliance.

The most common and effective way to fabricate aligned electrospun nanofibers is the use of a rotating collector (Figure 5B). A general mainstay of electrospinning on a rotating collector is that the degree of alignment and the diameter of the nanofibers could be controlled by controlling the rotating speed of the collector. High rotating speed of the mandrel results in enhanced stretching of the fibers, which gives higher fiber alignment and smaller fibers. Co-cultured hMSCs with HUVECs on aligned and random PCL fibrous scaffolds were investigated to study how fiber alignment modulates the osteogenic differentiation of hMSCs. The results demonstrated that aligned PCL fibers strongly influenced the morphologies and orientations of hMSCs and HUVECs. Aligned fibers could be, therefore, introduced to regenerate bone tissues with oriented topography without significant deleterious effects on hMSCs differentiation [111]. 

### 3.3. Honeycomb Nanofibers

A honeycomb is composed of hexagonal wax cells constructed by honeybees in their nests [115]. The natural honeycomb structure shows good features for cell culture, including a large surface area, good stability, and good permeability [116]. In recent years, many researchers have showed interest in making ordered honeycomb scaffolds which mimic nature’s honeycomb structure. Garcia et al. created a PCL and hydroxyapatite (HA) honeycomb scaffold, which could facilitate osteocompatibility in vitro and osteoinduction in vivo [117]. 

To fabricate honeycomb patterns by electrospinning, two methods have proven successful: (1) self-assembly [118] and (2) a collector equipped with hexagonal micropatterns [117]. The self-assembly approach is an easy, fast, and effective way to obtain honeycomb patterns. Honeycomb nanofibrous scaffolds have been successfully fabricated in our laboratory by self-assembly (Figure 5C). By changing the electrospinning parameters, such as the polymer concentration, peak voltage, and collecting distance, we could produce honeycomb nanofibrous scaffolds with different diameters [119]. HUVECs have also been cultured on honeycomb scaffolds as a biocompatible template to induce and regulate the formation of vascular network. Results showed that the honeycomb patterned nanofibrous scaffolds could support endothelial cell adhesion, proliferation, and migration into the scaffolds. Moreover, HUVECs cultured on the honeycomb scaffolds reorganized their cell bodies into tube-like structures, suggesting that HUVEC morphogenesis can be regulated by geometrical cues.

In addition to honeycomb patterns, other studies reported achieving fiber patterns through directed fiber deposition, such as in the case of the direct writing approach [120,121]. Lee and his colleagues made concentric circles and lattice patterned PCL electrospun fibrous mesh by using this approach [121]. The direct writing approach requires (1) a side-wall electrode, which is placed around the electrospinning jet to focus the fiber deposition, and (2) a X-Y stage to control the pattern during writing. The patterned nanofibrous mesh could be used for cell patterning in tissue engineering. We previously developed complex 3D poly(ethylene oxide terephthalate)/poly-(butylene terephthalate) (PEOT/PBT) fibrous scaffolds with articular cartilage mimetic structure by the direct writing method [122]. The effect of mimetic pattern on chondrogenic differentiation was also studied, demonstrating that patterned scaffolds guided fibril matrix organization and promoted the expression of chondrogenic markers (Sox9 and aggrecan). As human tissues always exhibit unique anisotropic structures, these results showed the great potential of using patterned fibrous structures (like in the case of honeycomb and direct writing methods here discussed) to generate complex tissue mimetic patterns, which could be used for inducing tissue regeneration. 

### 3.4. Spider Webs

A conventional electrospun membrane only produces one type of (micro or nano) fiber. An improvement of recent electrospinning approaches is creating complex electrospun scaffolds with both micro and nanofibers at the same time. Electrospun fibers in the nano scale could mimic native ECM fibers (a diameter in the range of 10–50 nm), while microfibers create highly porous scaffolds which benefit cell infiltration and migration. The fabrication of electrospun scaffolds that contain spider-web-like nano-nets with two types of (micro or nano) fibers could enhance scaffolds properties for tissue engineering applications [123] and could improve the mechanical properties of scaffolds. 

Nylon-6 and poly (acrylic acid) nano-nets were first reported, which were fabricated by tuning electrospun parameters [124]. Four years later, Pant et al. reported another spider-web-like nanofiber mesh, which contained thin methoxy poly(ethylene glycol) (MPEG) nanofibers and thick nylon-6 nanofibers. Those composite nanofibrous mats could increase mechanical strength and hydrophilicity compared with neat nylon-6 nanofibers (Figure 5D) [125]. Scaffolds with 1 wt % MPEG nanofibers showed higher mechanical strength because of the interconnected MPEG web-nanofiber with nylon-6 thick nanofibers. Tiwari and their colleagues fabricated a PCL-human serum albumin (PCL-HSA) membrane with spider-web-like nano-nets by electrospinning a blended solution of PCL and HSA [126]. The phase separation during electrospinning led to the formation of two types of fibers with different diameters. They also evaluated cell behavior on PCL-HSA scaffolds and proved that the membrane containing HSA nano-nets had better support for cellular activities compared to PCL scaffolds. Another study developed a polycaprolactone/polyethylene glycol (PCL/PEG) electrospun fibrous mesh with embedded nano-nets [127]. The stress-strain curves showed that the addition of PEG nano-nets could increase the tensile stress of PCL fibers from 13.40 ± 0.9 MPa to 27.20 ± 1.4 MPa. Higher mechanical strength of PCL/PEG nanofibrous mesh benefits cell attachment and growth. 

Dual air jet spinning (AJS) is another strategy with which to fabricate composite electrospun fibers with different diameters. The use of dual AJS could simultaneously control the mixing of two different polymers fibers from two AJS nozzles. This platform could afford the possibility of mixing two polymers together to produce web-like fibers with different diameters. Abdal-hay et al. reported that composite fibrous scaffolds composed of PCL and polyamide-6 (PA-6) fibers were successfully fabricated by using the dual AJS technique. Fiber diameter distribution showed that the diameters of PA-6 fibers were 278 ± 54 nm, while pure PCL had thick fibers with diameters of 2345 ± 152 nm. The composite fibrous scaffolds had better tensile properties than pure fibrous scaffolds. In vitro study demonstrated the PCL/PA-6 composite fibrous scaffolds provide a favorable microenvironment that supports attachment and proliferation of endothelial cells, so they could be promising scaffolds for vascular tissue engineering. Other studies have also shown spider-web-like structures in different electrospun polymeric membranes can better mimic the ECM and displayed enhanced mechanical properties, as in the cases here discussed. These scaffolds could be especially beneficial for cell attachment and migration in comparison to membranes without nano-webs [127,128,129]. A potential limitation of nano-web scaffolds is that the addition of the nanofibers on the microfibers might interfere with cell infiltration and penetration into microfibrous scaffolds.

### 3.5. Curly Fibers

Inspired by the structures of proteins in different tissues of our body, nanofibers with curved morphologies display many specific features, including large porosity, elasticity, and toughness, which make it attractive in biomedical tissue engineering. Three techniques has been reported to produce waved nanofibers: (1) self-assembly [130,131], (2) chemical vapor deposition [132], and (3) electrospinning [133,134,135]. Wang et al. reported that two synthesized β-sheet dipeptides with long alkyl chains derived from l-threonine and *D*-threonine (Thr) and sodium hydroxide (NaOH) could self-assemble into helical nanofibers [136]. Peptides amphiphiles with longer peptide sequences have also been reported to form helical nanofibers, due to the hydrogen bonding interactions between side chains inducing the formation of twisted nanofibers. Chemical vapor deposition was considered to be another technique generating helical fibers. Straight nanowires have been established to grow by the mechanism of vapor–liquid–solid (V–L–S). The adsorption of a gas phase deposit on to a solid surface leads to the growth of crystal from bottom to top, thereby creating the nanofibrillar structures. Zhang and his team produced helical SiO_2_ nanofibers using that technique, with diameters of the fabricated nanofibers from 80 to 140 nm [132]. Although the addition of CH_4_ flow in the carrier gas resulted in a helical structure of the formed nanofibers, the actual mechanism with which a helical structure was formed was not reported. The researchers hypothesized that CH_4_ helped with moderating the highly oxidizing environment and retained the activity of iron-catalyzed particles. 

Electrospinning has been regarded as the most common and efficient technique with which to fabricate helical nanofibers. Researchers produced helical nanofibers by electrospinning and studied the potential factors that could guide the fibers to be helical and curly. Zhou et al. created wavy nanofibers through an airflow-electrospinning process [137]. Airflow could drive electrospun nanofibers to deposit into a hybrid yarn, and then form wavy structures. Another study demonstrated that using auxiliary electrodes in near-field electrospinning could regulate the deposited fibers to form a wavy pattern [138]. They found that the frequency of the wavy fibers was the same as the current power frequency. We investigated the use of a thermally shrinking film to guide electrospun fibers in the formation of curly and wavy patterns (Figure 5E) [113]. In vitro studies showed that wavy patterns could enhance cellular infiltration and induce transforming growth factor-beta (TGF-β) expression. These hierarchically structured nanofibers could be used to mimic epithelial tissue buckling morphogenesis for tissue formation. The wavy pattern could afford unique physical properties to regulate cell behavior, which showed the possibility for application in tissue engineering. 

### 3.6. Nanotubes

Tubular nanostructures are widely used in tissue engineering, especially for drug delivery [139]. Many approaches have been studied to produce nanotubes for drug delivery systems. For instance, nanotube structures could be developed from organic building blocks by the self-assembly approach [140,141,142]. Yan et al. demonstrated that cationic dipeptides (H-Phe-Phe-NH_2_⋅HCl) could form nanotubes at physiological pH via self-assembly methods. To test the delivery of these cationic dipeptide nanotubes in cells, fluorescein labeled single stranded DNA (ssDNA) was bound to the nanotubes by electrostatic interactions. After incubation with HeLa cells, the fluorescence showed that these nanotubes could enter cells and accumulate in the cytoplasm. Nanotubes are also promising for potential applications of gene and drug delivery. However, low yield is a significant limitation to produce nanotubes in a large scale. The template-directed approach has also been used to build nanotubes, since it is more direct and controllable compared to the self-assembly approach [143,144]. Hou et al. produced poly(p-xylylene) nanotubes by this approach [144]. The average inner tube diameter was in the range of about 10 nm. The template approach works well for producing short tubes, but has difficulties fabricating continuous nanotubes. 

Compared with self-assembly and template-directed approaches, coaxial electrospinning was proved to be the optimal approach for creating continuous nanotubes, with controllable conditions and large-scale production. In co-electrospinning, a coaxial nozzle is the key device that is comprised of a central tube and a concentric circular corona. This coaxial nozzle allows two different solutions to produce a core-shell composite structure [145,146]. In order to fabricate hollow nanofibers, it was shown that different oil-soluble materials could be directly spun into the core of the fiber [114,147]. After removal of the oil phase from the core of the scaffolds, hollow nanofibers were obtained (Figure 5F) [114]. Srouji et al. developed electrospun core-shell fiber scaffolds to load bone morphogenetic protein-2 (BMP-2) to the cores of the fibers [148]. After BMP-2’s slow release from the scaffolds, both in vitro and vivo studies showed that slow-releasing scaffolds have significant increases in alkaline phosphatase activity and bone regeneration compared to fast BMP-2 releasing scaffolds. Moreover, Zhao et al. successfully developed multi-channel tubes with variable diameters and channel numbers. The channel number of nanotubes was controlled by a powerful multifluidic compound-jet during electrospinning [149]. Multi-channel tubes could be used to load different bioactive components without interaction in drug release. Core-shell electrospinning serves as an ideal approach to load bioactive molecules for tissue engineering applications, which will be further discussed in the next section.

Currently, nanofibers with different patterns have been successfully fabricated by adjusting the processing parameters of electrospinning. Nanofibers could be designed as specific structures depending on the biomedical applications, due to every structure having its own advantages. For example, aligned fibers were used to better mimic the collagen fibers in bone matrix [104]. Honeycomb patterns mimicking the capillary network structure of endothelial cells could be promising for vascular tissue regeneration. Core-shell fibers could be used as delivery carriers in which drugs or bioactive components are encapsulated in the shell and achieve sustainable release [139]. Spider-web-like structures can better mimic the ECM with different fiber sizes and display-enhanced mechanical properties [127,128,129]. Wavy patterns could be used to mimic tissue buckling morphogenesis for tissue formation [113]. Therefore, investigating the structural and mechanical properties of different tissues is important in order to choose the specific structure that is best for the regenerative application. 

## 4. Nanofiber Functionalization

To explore more applications of nanofibrous scaffolds in tissue engineering, researchers investigated the functionalization of nanofibrous scaffolds by many different approaches, such as physical blends, core-shell electrospinning, and post functionalization (Figure 6). Physical blends and core-shell electrospinning could directly fabricate functional nanofibers during the electrospun process. Post functionalization requires relatively stricter conditions to ensure the functional molecules could successfully bind to nanofibers without affecting their structure. Some of the advantages and disadvantages of each method are listed in Table 2.

### 4.1. Physical Blends

Using polymer blends is an easy and direct method to introduce bioactive components into nanofibrous scaffolds for tissue-engineering applications. Nanostructure polymer blends could result in the production of low-cost scaffolds with specific functionality. Biopolymers can be blended into a synthetic polymer for electrospinning to obtain better mechanical properties, good biodegradability, and good biocompatibility of nanofibers. For example, to improve the biodegradability and biocompatibility, gelatin was mixed into a PCL electrospun solution and then incorporated into PCL nanofibrous scaffolds during electrospinning [150]. The obtained PCL/gelatin nanofibrous scaffolds exerted positive influences on cell proliferation and nerve differentiation compared with pure PCL scaffolds. Numerous natural polymers have been blended with synthetic polymers for electrospinning, such as collagen [151,152,153], gelatin [154,155,156], elastin [157,158,159], fibrinogen [160], silk fibroin [161], alginate [162,163], chitosan [164,165], dextran [166,167], and heparin [168]. These results highlight the essential role of both natural and synthetic polymers in tissue engineering applications. 

The combination of natural and synthetic polymers could effectively improve the mechanical properties and biocompatibility of nanofibrous scaffolds, which could support cell growth and favor tissue regeneration. More importantly, the simple physical blending process for the introduction of functional components to polymeric nanofibers usually resulted in converting the nanofibers into highly effective delivery systems. Many biofunctional components (such as growth factors [169,170], peptides [171,172], enzymes [173,174], drugs [175,176], and DNA [177,178]) have been incorporated into nanofibers for biomedical research. After incorporating bioactive components into polymer nanofibers, the composite scaffolds could serve as carriers to provide continued or controlled release of the bioactive components for tissue regeneration. Selecting polymers (such as biodegradable and non-degradable) as carriers could offer the possibility of controlling the speed of drug release. By selecting different polymers with a variety of biofunctional components, nanofibrous scaffolds provide high versatility for different applications.

### 4.2. Core-Shell Electrospinning

Electrospinning is an effective technique with which to incorporate biomolecules into nanofibrous scaffolds. However, physical blends provide few handles to tune the release of bioactive molecules, and generally create relatively fast release compared with coaxial electrospinning. The burst release often results in a short lifetime of the bioactive scaffolds and reduced therapeutic efficiency of bioactive molecules. Coaxial electrospinning has been designed to produce core-shell nanofibers, which facilitate the biomolecules encapsulation into the core of nanofibers. This specific coaxial structure is essential to avoid burst release, slow down the release rate, and maintain the mechanical and biological properties of scaffolds. 

Researchers compared the loading efficiency and releasing properties of two methods (physical blends and coaxial electrospinning) [169]. They fabricated two different poly(lactic-co-glycolic acid) (PLGA) nanofiber scaffolds encapsulating basic fibroblast growth factor (bFGF) by using physical blends and coaxial electrospinning. Although both PLGA nanofibers showed growth factor encapsulation and achieved continued release, the release profiles were quite different. Co-electrospinning scaffolds could prolong the bFGF release one week more compared to physical blends scaffolds. Therefore, coaxial electrospinning is more effective at loading growth factors and achieving continued release. Zhang et al. encapsulated vascular endothelial growth factor (VEGF) and platelet-derived growth factor-bb (PDGF) into fibrous meshes via coaxial electrospinning technique. VEGF was loaded in a chitosan hydrogel/poly(ethylene glycol)-b-poly(l-lactide-co-caprolactone) (PELCL) electrospun membrane as the inner layer and PDGF was loaded in an emulsion/PELCL electrospun membrane as the outer layer. Results demonstrated that the release of those two growth factors at same time could promote vascular endothelial cell (VEC) proliferation and facilitate revascularization [179]. Mickova and his colleagues designed polyvinyl alcohol/PCL as core/shell nanofibers with embedded liposomes in the core. Results showed that the enzymatic activity of encapsulated horseradish peroxidase was still maintained in the scaffolds, which could be used in a drug delivery system [180]. In addition to growth factors and liposomes, controlled release of other bioactive components, such as peptides, proteins, and even DNA, is also accomplished with core-shell electrospinning approaches [181]. 

### 4.3. Post Functionalization

#### 4.3.1. Plasma Treatments

Simple plasma treatment is a widely used method to improve nanofiber biocompatibility. Different gases such as Ar, N_2_, O_2_, and NH_3_ have been used during plasma treatment to change the chemical composition or introduce functional groups on the surfaces of scaffolds for further applications. For example, polymer nanofibers were exposed with oxygen plasma in order to obtain carboxyl groups on the surface of nanofibers [182]. Plasma treatments with ammonia can generate amine active sites on the surface of nanofibers [183]. After generating desired functional groups (e.g., –H_2_, –OH, and –COOH) on the surfaces of nanofibers, these surfaces could be largely used as substrates for biomolecule immobilization. 

A study demonstrated the functionalization of PLLA nanofibers with RGD (Arg–Gly–Asp) peptides by plasma treatments and covalent coupling. The plasma treatment of PLLA nanofibers is a key step to producing carboxyl groups for RGD coupling. RGD peptides could immobilize on the functionalized nanofibers by 1-Ethyl-3-(3-dimethylaminopropyl) carbodiimide/N-hydroxy succinimide (EDC/NHS) chemistry. In vitro results showed that RGD functionalized scaffolds promoted the proliferation and osteogenic differentiation of hMSCs [184]. Another study functionalized PLLA nanofibers with cationized gelatin (CG) to increase the biocompatibility of PLLA scaffolds [182]. Oxygen plasma was used to modify PLLA nanofibers with carboxyl groups. After the introduction of carboxyl groups on the surface of nanofibers, gelatin could covalently bind to the PLLA nanofibers. Apart from growth factors and gelatin, other bioactive components (calcium phosphate [185], collagen [186], and laminin [187]) were also reported to bind onto nanofibers after plasma treatment. Plasma treatment was regarded as a simple approach to generate functional groups on the surface, which provides an opportunity for the further immobilization of bioactive components onto scaffolds.

#### 4.3.2. Wet Chemical Etching

The surface modification of nanofibrous scaffolds could also be achieved by wet chemical treatments. Wet chemical approaches, such as hydrolysis, and soaking in NaOH or KOH, often lead to the degradation of nanofiber properties and morphology. Chen et al. reported that PCL nanofibrous membranes treated with NaOH etching led to enhanced roughness and surface area [188]. 3T3 fibroblasts exhibited a better adhesion on the surface modified PCL membranes and well spread-out morphology. Other chemical etching approaches, such as aminolysis, could introduce amino groups onto the nanofiber surface. Amino modified nanofibers could be used as a substrate to bind other bioactive molecules by a covalent bond with the –COOH group. Fibronectin (Fn), for example, was grafted onto nanofibrous scaffolds via polyester aminolysis and then coupling via glutaraldehyde [189]. Cell studies demonstrated that the Fn coupling onto polymer nanofiber scaffolds significantly promoted epithelium regeneration. The introduction of active groups by using wet chemical etching affords an easy route for preparing functional nanofibrous materials in tissue engineering. A variety of bioactive components, such as growth factors, peptides, gelatin, and chitosan, can be functionalized into scaffolds afterwards. Therefore, this functionalization approach showed high potential for tissue regeneration applications.

#### 4.3.3. Click Chemistry

Click chemistry is an attractive alternative for facile immobilization of bioactive components onto electrospun nanofiber surface, because of its simple reaction conditions, high reaction rate, high yields, and functional group tolerance [190,191,192,193]. Several “click” methods, including radical thiol-ene [194], copper-catalyzed azide alkyne cycloaddition (CuAAC) [195,196], strain-promoted azide–alkyne cycloaddition (SPAAC) [197,198], Diels−Alder addition [199], and Michael additions [200] have been employed for nanofiber functionalization [201]. Modification of nanofibers surface with bioactive components by click chemistry could be used to improve the bio-functional properties of nanofibrous scaffolds for tissue regeneration applications. For example, Lancuški et al. developed clickable nanofibrous scaffolds by electrospinning a mixture of PCL-80K with PCL-2K azide. The surface of PCL-2K azide presented clickable sites for CuAAc reaction. This approach provides high efficiency with which to produce clickable nanofibrous scaffolds [195]. Then, functionalized nanofibers could easily click to desired biomolecules (peptides, growth factor, drugs, etc.). Another study showed a breast cancer sensitive protease (TSP50) was successfully coupled onto P(LA90-co-MPC10) electrospun fibers by CuAAc click reaction [196]. The TSP50 fibers could be applied for detection, separation, and purification of anti-TSP50. This method verifies the possibility using a click reaction to couple proteins on electrospun fibers. Photo-initiated thiol-ene click chemistry is also widely used to bind bioactive molecules. For example, a cysteine contained Cys-Ala-Gly (CAG) peptide was functionalized onto scaffolds via thiol-ene click chemistry [202]. These peptide-functionalized surfaces exhibited enhanced cell adhesion, growth, and proliferation. Compared to previous chemical functionalization of nanofibers, such as plasma treatment and the wet chemical method, click reactions are unique and offer many benefits (stoichiometric control, orthogonal chemistry, and mild water-tolerant conditions), which could effectively attach biomolecules to the surface of nanofibers without damaging a fibrous structure. As the control and use of click chemistry keeps increasing, clickable nanofibers have shown great promise to bond various biomolecules with different approaches for tissue engineering applications.

#### 4.3.4. Surface Graft

The surface graft approach has been also developed to introduce functional groups on scaffolds for the subsequent coupling of bioactive components. Plasma treatment is usually required to introduce free radicals for polymerization during graft polymerization. For example, the surface of poly(vinylidene) fluoride nanofibers was successfully grafted with poly(methyl methacrylate) by plasma-induced graft copolymerization [203]. By processing with graft polymerization, the pore size of electrospun nanofibrous meshes was significantly decreased, from 3.58 to 0.88 μm. The grafted electrospun membranes could be used as a microfiltration membrane. Ma et al. modified electrospun PCL nanofibers with gelatin by surface grafting [204]. Gelatin modified PCL nanofibrous scaffolds effectively helped endothelial cell (EC) adhesion and promoted proliferation compared with the unmodified PCL scaffolds. However, the grafting approach can only modify the top surfaces of nanofibers and could not penetrate beyond the first few fiber layers. 

## 5. Electrospun Fibrous Scaffolds in Vascular Regeneration

Vascular regeneration is one of the most challenging issues in tissue engineering [205,206]. During formation of new tissue, blood vessels are required to supply oxygen and nutrition for cells, and remove waste products [207,208,209]. Lack of efficient vascularization limits the size of tissue-engineered constructs [210,211]. Implantation of tissue constructs in a poorly vascularized site often leads to lack of tissue integration and cell death [212,213]. As a result, many tissue-engineered constructs have been reported to fail in vivo due to the lack of vascular network formation [210,214,215]. Therefore, vascular regeneration is critical for the successful regeneration of tissues where vascularization is necessary [216,217]. Electrospinning is a commonly used technique to produce fibers at the micro/nano scale, which could mimic the mechanical properties of native ECM. The potential in vivo use of implantable electrospun tubular scaffolds for vascular graft was widely reported [218,219,220]. In particular, tubular electrospun scaffolds collected on a rotating mandrel showed advantages when mimicking the scale and architecture of vessels.

### 5.1. Effects of Electrospun Scaffold Architecture on Vascular Regeneration

Nanotopography has been noted as an important factor with which to affect cell growth and differentiation. ECs could interact with their physical environments and be guided by a scaffold’s topography. Xu et al. reported that the alignment of nanofibers plays a positive role in the regulation of endothelial cellular behavior [221]. Cell elongation and migration are indispensable processes in angiogenesis. This study proved that the alignment of nanofibers could guide cell distribution, affect cell morphology, and even control migration velocity [222]. When HUVECs were cultured on PLGA-aligned nanofibrous scaffolds, the morphologies of migrating cells were highly ordered. Therefore, the topographic features and scaffold guidance should be evaluated when designing a tissue-engineered scaffold for vascular regeneration. 

Apart from electrospun scaffolds, a few other studies showed that EC morphogenesis into capillary-like structures was regulated by micropatterned stripe substrates [217,223]. Dike et al. showed that ECs cultured on substrates micropatterned with 10 µm wide lines of fibronectin formed capillary tube-like structures containing a central lumen; cells cultured on wider (30 µm) lines did not form tubes [223]. Moon et al. micropatterned poly (ethylene glycol) diacrylate hydrogels with RGDS in different geometries [217]. As a result, ECs cultured on RGDS patterns reorganized their cell bodies into tube-like structures on 50 µm wide stripes, but not on wider stripes. These results suggested that EC morphogenesis could be regulated by topography cues. The development of a well-designed topography, in which capillary tubes consistently form, is an important step toward the fabrication of engineered tissues.

### 5.2. Effects of the Controlled Release of Biochemicals from Electrospun Scaffolds on Vascular Regeneration

#### 5.2.1. Growth Factors

Many researchers have reported that the controlled release of angiogenic factors could promote vascular formation in vitro and in vivo. Electrospun nanofibrous scaffolds show potential to incorporate or functionalize bioactive components onto the scaffolds, and scaffolds designed for drug release have shown controlled delivery. VEGF is the most important growth factor to stimulate early vascular formation and promote angiogenesis. Jia and his colleagues loaded VEGF into the inner of core/shell fibrous scaffold by coaxial electrospinning with PLGA as the shell [224]. VEGF release could be sustained for more than 28 days and cell studies showed that VEGF encapsulated scaffolds effectively enhanced cell proliferation and benefited cell distribution. Another study demonstrated VEGF functionalized heparin-conjugated PCL fibrous scaffolds were able to release the growth factor for 15 days [225]. This resulted in new blood vessel formation with minimum immunological rejection. Other growth factors, such as bFGF [226] and PDGF [179], were also reported to be released from nanofibrous scaffolds for vascular regeneration. Montero et al. prepared bFGF-loaded gelatin fibrous scaffolds by physical adsorption after electrospinning [227]. HUVECs were seeded on these scaffolds. Results showed that the releasing of bFGF from the scaffolds significantly promoted cell proliferation and helped capillary formation. Moreover, combing two or more growth factors and controlling their spatio-temporal release could be another option for improving the functionality of scaffolds. 

#### 5.2.2. VEGF-Mimetic Peptides

Many studies have also reported the immobilization of VEGF-mimetic peptides on scaffolds as a potential solution for vascular regeneration. D’Andrea et al. designed a VEGF-mimetic peptide, QK (domain: KLTWQELYQLKYKGI), and showed the ability to activate VEGF receptors and similar bioactivity to VEGF [228]. QK peptides provided many advantages, including low molecular weight, low immunogenic potential, and cost-effectiveness by synthesis [228,229,230,231,232]. Leslie-Barbick et al. reported that QK peptides were easier to conjugate or immobilize into scaffolds than VEGF because they could diffuse into scaffolds faster and more completely [232]. Another study loaded QK peptide into poly(ethylene glycol)-b-poly(L-lactide-co-e-caprolactone) (PELCL) nanofibers by emulsion or suspension electrospinning [233]. It was found that QK loaded PELCL electrospun scaffolds could significantly accelerate the proliferation of ECs compared with pure PELCL scaffolds in nine days. Zhou et al. functionalized the surface of a PELCL scaffold with QK peptides via EDC/NHS chemistry [231]. In vitro studies demonstrated that the QK peptide-functionalized PELCL scaffolds could significantly promote the proliferation of ECs compared with unfunctionalized PELCL scaffolds. QK peptide-functionalized electrospun scaffolds showed their ability of fast endothelialization, which could have potential use in vascular regeneration. 

Apart from VEGF-memetic peptides, RGD peptide was also reported to immobilize into electrospun scaffolds for vascular tissue engineering. For example, Kim et al electrospun a mixture of PLGA and PLGA-b-PEG-NH_2_ to generate electrospun scaffolds with a functionalizable amine [234]. RGD was covalently grafted with on PLGA fibrous scaffolds. In vitro study showed that the immobilization of RGD significantly promoted cell adhesion and proliferation. These results suggest that RGD functionalized fibrous scaffolds could be promising for vascular regeneration.

#### 5.2.3. Hydrogen Sulfide

Hydrogen sulfide (H_2_S), a unique gasotransmitter that has been recognized as an important physiological and pathological signaling molecule, can mediate and promote the effects on angiogenesis [235,236]. The phenomenon that H_2_S promotes EC proliferation and migration has been reported by different groups [237,238]. H_2_S has been reported to simulate angiogenesis in vitro and in vivo [237,239]. Since H_2_S has been recognized to be beneficial for angiogenesis, researchers started to focus on the development of H_2_S releasing scaffolds for vascular regeneration [240,241]. Feng et al. electrospun N-(benzoylthio)benzamide (NSHD1), an H_2_S donor, with PCL solution to form H_2_S release fibrous scaffolds [242]. The H_2_S fibrous scaffolds could facilitate (H9c2 and 3T3) cell proliferation. Moreover, these scaffolds were also reported to increase the expression of collagen type I and collagen type III, and wound healing related genes. Kang and his colleagues synthesized a pH-controlled H_2_S donor (JK-1) from phenylphosphonothioic dichloride [243]. Wu et al., instead, mixed JK-1 with PCL to prepare H_2_S releasing electrospun nanofibers [244]. The fabricated fibrous scaffolds showed that lower pH induced greater and faster H_2_S release. JK1 doped PCL fibers were nontoxic to fibroblasts and in vivo experiments proved that PCL-JK1 could significantly improve wound healing. Although previous studies showed promising results in vitro and in vivo, many aspects still need to be improved for the fabrication of H_2_S releasing nanofibrous scaffolds: (1) a slow H_2_S-releasing donor needs to be included; (2) in addition to physical doping, other surface functionalization approaches should be considered. 

### 5.3. Studying Angiogenesis In Vivo

The chick chorioallantoic membrane (CAM) assay and rabbit or rat corneas are the most widely used animal models for studying the process of angiogenesis in vivo. CAM is an extraembryonic membrane mediating gas and nutrient exchanges until hatching [245]. Since this membrane could form blood vessels network after incubation, it has been widely performed as an in vivo model to screen angiogenesis stimulators and inhibitors in response to biomaterials. Rabbit cornea has been reported to be another in vivo angiogenesis model. However, CAM is believed to be simpler and more cost-effective with lower ethical concerns than other animal models. 

CAM assay has been used by many researchers as an in vivo model in vascular tissue engineering for more than 40 years. A number of studies have evaluated electrospun nanofibrous scaffolds on the CAM to examine their angiogenic response and biocompatibility. The general procedure of CAM assay is shown in Figure 7. By implanting the scaffolds onto the CAM, their potential angiogenic activities can be analyzed through changes in the vascular density of the surrounding environment. Test components can be growth factors, peptides, drugs, and other biomolecules, which are immobilized or incorporated on nanofibrous scaffolds. For example, in order to evaluate the angiogenic ability of VEGF-loaded collagen-PCL scaffolds in vivo, a CAM assay was carried out, showing that VEGF-loaded PCL scaffolds could significantly increase the vessel area on the scaffolds [246] Their result proved that the VEGF released from VEGF-loaded scaffolds could promote early blood vessel formation in vivo. Augustine et al. developed zinc oxide (ZnO) nanoparticle-loaded electrospun PCL scaffolds [247]. A 1 wt% ZnO nanoparticle incorporated PCL scaffold was pro-angiogenic, and it proved to have statistically more branching points than PCL scaffolds in a CAM assay. Diaz-Gomez et al. performed a CAM assay to evaluate the angiogenesis response of platelet-rich plasma (PRP)-coated PCL scaffolds [248]. PRP-PCL scaffolds were integrated with the CAM and formed many capillary blood vessels around the PRP-PCL scaffolds compared to the PCL scaffolds. These few selected studies are just to exemplify the potential of the CAM assay as a simple validation tool for newly designed scaffolds for vascular tissue regeneration, after an initial assessment *in vitro*. For a more thorough review on the several angiogenesis available assays, the reader is referred elsewhere [249]. 

## 6. Future Outlook and Conclusions

Scaffolds for tissue engineering have become more and more important in regenerative medicine, which is crucial for the development of advanced clinical therapy for those critical tissue injuries that do not heal correctly by themselves [250]. The key factors for tissue engineering include cells, biological factors, and scaffolds, which are used alone or in combination to promote and guide cell proliferation and differentiation, and consequently tissue regeneration [2]. In recent years, many studies have focused on the development of novel combinatorial approaches to design bioactive scaffolds for tissue regeneration. To reach these goals, electrospinning is often used, since it is a simple and promising technique for the fabrication of biomimetic scaffolds that are capable of mimicking the physical properties and structure of the natural extracellular matrix (ECM) [251]. The properties of electrospun nanofiber scaffolds, especially their topographical features, can be easily controlled by adjusting electrospinning parameters. In addition, electrospun fibrous scaffolds functionalized with bioactive components have been widely used to promote cell repair and tissue regeneration. However, there are still some challenges remaining for electrospun scaffolds. 

(1) Structure design.

Most tissue regeneration, such as that of bone, muscle, and cartilage, necessitates 3D scaffolds, but nanofibrous scaffolds with high thickness are difficult to obtain by electrospinning. Electrospinning normally produces 2D random nanofibrous scaffolds with high porosity in the height of few microns [252]. Honeycomb nanofibrous scaffolds were fabricated with higher depth, reaching 130 µm, which is still at the microscale range [119]. Therefore, there is high demand to explore more microfabrication approaches combined with electrospinning to increase the thicknesses of nanofibrous scaffolds. Combining electrospinning with melt electrospinning or additive manufacturing might be a possible approach with which to generate electrospun fibers with greater thicknesses. 3D filaments could act as a template layer to increase the depth of a nanofibrous mesh and also create high porosity. Porous nanofibrous scaffolds with greater depths provide common and versatile 3D environments for cell migration and infiltration, which could be an important step forward in the field of electrospun scaffolds for tissue regeneration applications.

(2) Nanofiber functionalization

Traditional nanofiber functionalization approaches lack of selective control. Ideally, functionalization could also be provided in specific locations in 3D scaffolds through the presence of selective reactive groups. For example, this could be reached by photopatterning and would allow for achieving spatial control of the fiber functionalization. The ability to use a photopatterning approach to create any spatial functionalization of fibrous scaffolds is a significant advantage, which could be used for the partial functionalization of bioactive molecules on fibrous scaffolds for vascular tissue regeneration. 

(3) Vascular tissue regeneration

An ideal scaffold should not only provide an appropriate physical environment [28,253], which mimics the mechanical and structural properties of the target tissue, but also provide growth factors or other bioactive molecules to guide cellular behavior [254,255]. This review only described different improved scaffolds based on structural design or nanofiber functionalization to enhance angiogenesis and the vascularization of tissues. In the future, a combination of these two factors, such as combining the honeycomb pattern with functionalization of angiogenic growth factors on nanofibrous scaffolds, may have a synergistic impact on directing cell organization towards tissue engineering of functional vascular networks. Only with tight control over multiple cues and rigorous testing in real situations (in vivo) will we be able to remove the vascularization bottleneck in tissue engineering.

## Figures and Tables

**Figure 1 nanomaterials-10-00887-f001:**
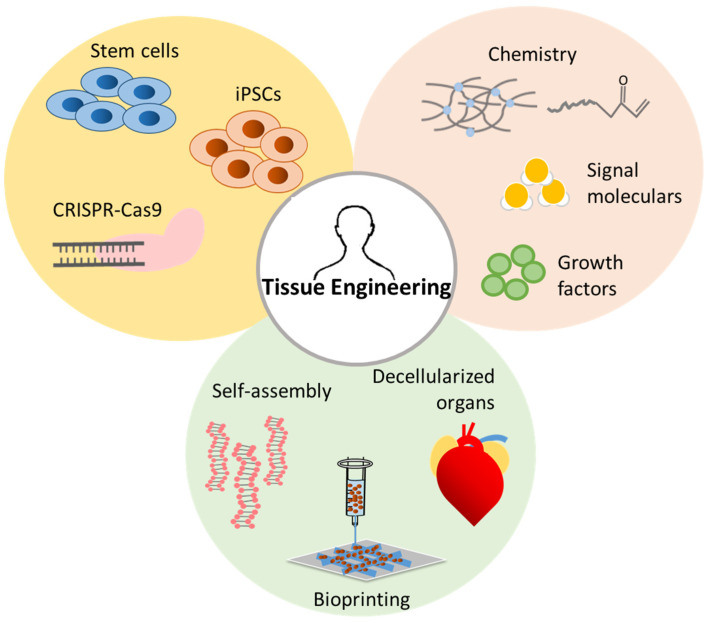
Summary of tissue engineering progress.

**Figure 2 nanomaterials-10-00887-f002:**
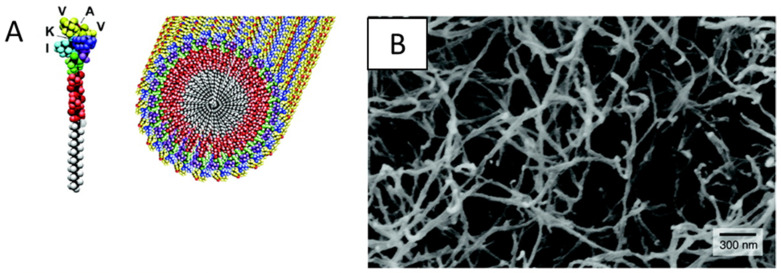
(**A**) Molecular model of a peptide amphiphiles (IKVAV) and (**B**) SEM images of a self-assembling peptide–amphiphile nanofiber network. Reproduced with permission from [64]. Copyright 2004, AAAS.

**Figure 3 nanomaterials-10-00887-f003:**
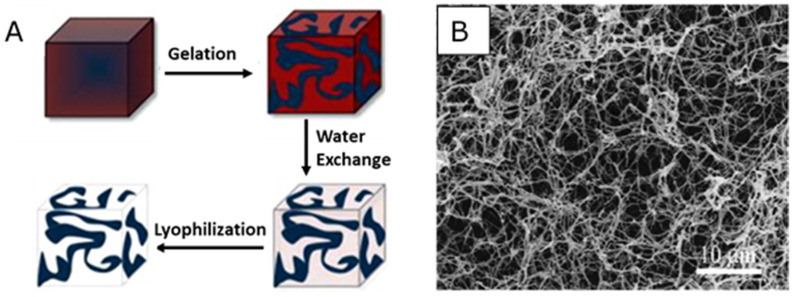
A schematic (**A**) of nanofiber formation by phase separation, and an SEM image (**B**) of nanofibrous structure fabricated by this technique. (**A**) was reproduced with permission from [66]. Copyright 2012, Elsevier Ltd. SEM image is reproduced with permission from [37]. Copyright 1999, John Wiley and Sons.

**Figure 4 nanomaterials-10-00887-f004:**
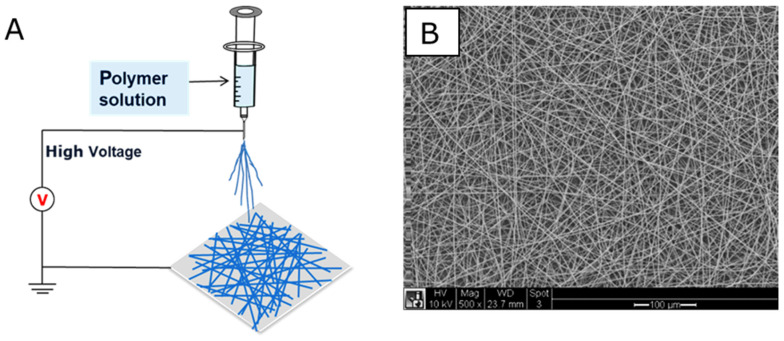
Schematic (**A**) of a typical electrospinning system, and SEM image; (**B**) of electrospun nanofibrous structure.

**Figure 5 nanomaterials-10-00887-f005:**
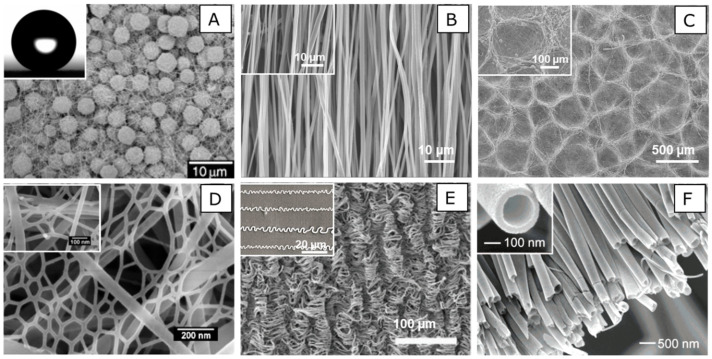
Different nanofibrous structures fabricated by electrospinning approach. SEM images of (**A**) a porous microsphere/nanofiber composite, (**B**) an aligned nanofiber, (**C**) a honeycomb nanofiber, (**D**) a spider-web-like nanofiber, (**E**) a curly nanofiber, and (**F**) a nanotube. Porous microsphere/nanofiber composite: reproduced with permission from [91]. Copyright 2004, WILEY-VCH. Spider-web-like nanofiber: reproduced with permission from [112]. Copyright 2011. Elsevier B.V. Curly nanofiber: reproduced with permission from [113]. Copyright 2019, Royal Society of Chemistry. Nanotube: reproduced with permission from [114]. Copyright 2004, American Chemical Society.

**Figure 6 nanomaterials-10-00887-f006:**
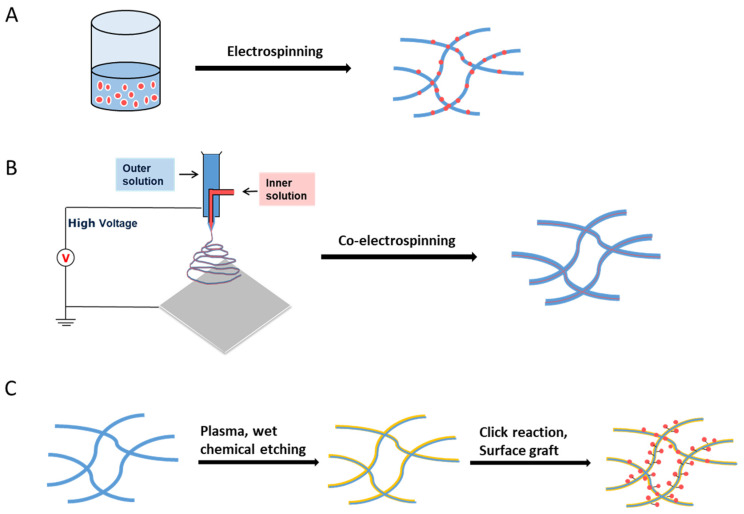
Functionalization approaches of electrospun nanofibers. (**A**) Physical blends, (**B**) core-shell electrospinning, and (**C**) post functionalization.

**Figure 7 nanomaterials-10-00887-f007:**
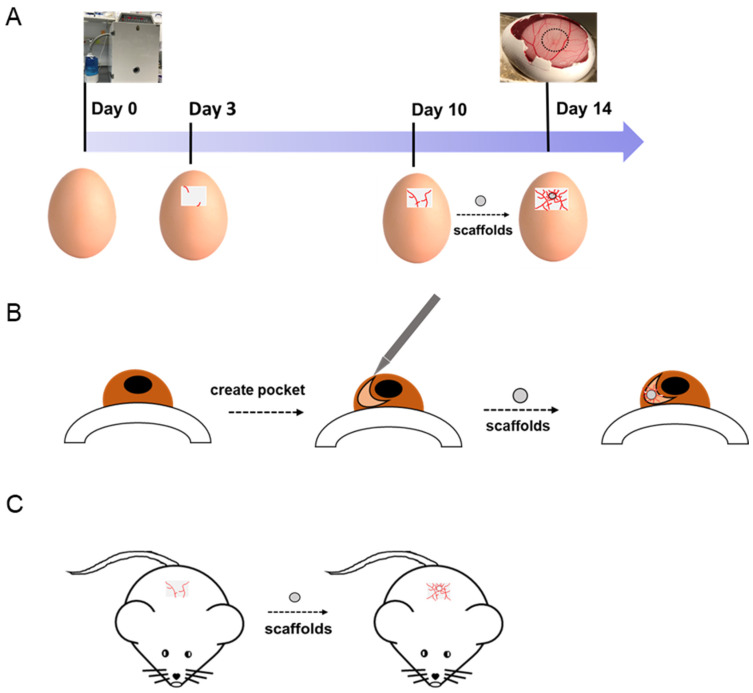
(**A**) Chick chorioallantoic membrane (CAM), (**B**) rabbit cornea, and (**C**) mouse models are used to evaluate the angiogenic abilities of implanted scaffolds.

**Table 1 nanomaterials-10-00887-t001:** Comparison of three different nanofiber fabrication methods.

Method	Advantages	Disadvantages
Self Assembly	Fibers at the nano scale Easy to obtain 3-dimensional (3D) porous structures Cells can be encapsulated during fiber formation Bioinspired Injectable solutions for minimally invasive applications	Complex process Poor control over fiber orientation Limited fiber diameter and length Mostly empirical control over structures.
Phase Separation	Easy to obtain 3D porous structures Tailorable mechanical properties	Complex process Lower control over fiber orientation
Electrospinning	Well-established Cost effective Easy to control fiber diameter, microstructure and arrangement Wide choice of biomaterials can be used	Poor cell infiltration and penetration into the scaffolds Lack of control over pore arrangement in 3D Potential toxicity of solvents used during electrospinning

**Table 2 nanomaterials-10-00887-t002:** Comparison of methods for nanofiber modification.

Method	Advantages	Disadvantages
Physical blends	Direct and easy Uniform functionalization	Burst release Only limited amount of functional group on the surface of nanofiber Functionality must survive processing
Core-Shell electrospinning	Functional components could be encapsulated in the core of fibers Sustained release	Complex process Strict conditions to produce core-shell structured nanofibers Functionality must survive processing
Post functionalization Plasma treatments Wet chemical etching Click chemistry Surface graft	Surface functionalization Many choice could be selected Tolerant to more sensitive functional molecules	Need to process after electrospinning Need relative strict condition without compromising nanofiber structure

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
