# Peer review of "Strategies to Improve Nanofibrous Scaffolds for Vascular Tissue Engineering"

_nanomaterials, 2020, doi:10.3390/nano10050887_

Round 1

Reviewer 1 Report

The manuscript review the literature on nanofibrous scaffolds for vascular tissue engineering. Specifically the authors describe the three major methods of producing nanofibers. They next focus on different design strategies, methods and design criteria for nanofiber fabrication. similarly, the authors discuss nanofiber functionalization techniques. Finally the focus on electrospun nanofibers for vascular tissue engineering.

The specific comments are below

In general the manuscript could be referenced better. There are several instances where the authors do not cite literature when necessary.

There are several instances where the transition between sections could be significantly improved and better definitions provided. There are clearly observable differences in style, depth and discussion. Thus, the lead/corresponding author should try to better homogenize the manuscript. For example, section 4.2 seems out of place as core-shell was discussed in 3.6. Similarly, it is unclear why hydrogels are described in section 5.1. Each paragraph/section should have pros/cons summary

Please provide a more in depth analysis there are many instances of vague descriptions. for example line 408 "physical blends cannot be effective for long time and often provide bioactive molecules with a fast release" - I am not sure this is correct. Similarly, line 452 "The simple plasma treatment is the most widely used method to improve nanofibers biocompatibility." the lines are sometimes blurred between what would be considered regenerative medicine vs tissue engineering and a better review of the literature is on vascular tissue engineering warranted. Often the authors discuss unrelated examples especially in bone tissue engineering  or drug delivery (not in line with title - perhaps authors should expand the scope?) Similarly, I am not sure blending is considered functinalization

There is a clear discrepancy when discussing their own work vs the work of others - their own work is very detailed whereas other examples read more anecdotal.

Define acronyms at first use such as EDC/NHS

There are several run on sentences (4-5 lines) or a one sentence paragraphs.

Should mention dual air jet spinning as a viable technique for vascular tissue engineering since it can simultaneously produce nano and micro fibers - section 3.4 example - see European Polymer Journal 96, 27-43

Author Response

Comments and Suggestions for Authors

The manuscript review the literature on nanofibrous scaffolds for vascular tissue engineering. Specifically the authors describe the three major methods of producing nanofibers. They next focus on different design strategies, methods and design criteria for nanofiber fabrication. similarly, the authors discuss nanofiber functionalization techniques. Finally the focus on electrospun nanofibers for vascular tissue engineering.

The specific comments are below

  1. In general the manuscript could be referenced better. There are several instances where the authors do not cite literature when necessary.

Reply. In section 4.2, we forget to cite reference in some sentences. Now already added.

  1. There are several instances where the transition between sections could be significantly improved and better definitions provided. There are clearly observable differences in style, depth and discussion. Thus, the lead/corresponding author should try to better homogenize the manuscript. For example, section 4.2 seems out of place as core-shell was discussed in 3.6. Similarly, it is unclear why hydrogels are described in section 5.1. Each paragraph/section should have pros/cons summary

Reply. We thank the reviewer for her/his critical view on our manuscript. In section 3.6, we introduced nanotubes in the structure design section, and we just mentioned core-shell electrospinning as one possible strategy to produce nanotubes. Other strategies, such as self-assemble approach, are also included in this section. In section 4.2, we focus on introducing the core-shell electrospinning, which have widely been used to incorporate biomolecules into nanofibers. In order to make it clear, we added a sentence in 3.6 saying “Core-shell electrospinning serves as an ideal approach to load bioactive molecules for tissue engineering applications, which will be further discussed in section 4.2”. Please check in 3.6.

In section 5.1, we described the effect of electrospun scaffold architecture on vascular regeneration. We also added some examples, which regulated endothelial cells to form tube-like structure by micropatterned stripe substrate (hydrogel). Although the examples provided are not related to electrospun scaffolds, these are still meaningful to prove that endothelial cells response to topographical cues of substrates can result in the formation of tube-like structures.

We added one sentence saying “Apart from electrospun scaffolds, few other studies showed that EC morphogenesis into capillary-like structures was regulated by micropatterned stripe substrates“. We hope it is clearer now.

We added a summary about advantages and disadvantages in every section. Please check tables 1 and 2, and section 3.

  1. Please provide a more in depth analysis there are many instances of vague descriptions. for example line 408 "physical blends cannot be effective for long time and often provide bioactive molecules with a fast release" - I am not sure this is correct. Similarly, line 452 "The simple plasma treatment is the most widely used method to improve nanofibers biocompatibility." the lines are sometimes blurred between what would be considered regenerative medicine vs tissue engineering and a better review of the literature is on vascular tissue engineering warranted. Often the authors discuss unrelated examples especially in bone tissue engineering  or drug delivery (not in line with title - perhaps authors should expand the scope?) Similarly, I am not sure blending is considered functionalization.

Reply. We have changed the sentence to“…physical blends cannot be effective for long time and often provide bioactive molecules with a relatively fast release compared with coaxial electrospinning…”. We wanted to compare the release from physical blends with co-electrospinning. The related reference that supported this sentence was also explained thereafter. Please check 4.2.

We have changed the sentence to “Simple plasma treatment is a widely used method to improve nanofibers biocompatibility”.

We mentioned bone tissue regeneration in 3.2 (aligned nanofibers), because there is a large amount of papers, which studied how aligned nanofibers could mimic the alignment of collagen fibers in bone matrix. Moreover, we also discussed the effect of aligned fibers on vascular tissue engineering in this section.

For physical blends, different bioactive components have often been incorporated into polymer scaffolds to add functionalities for tissue engineering applications. Although it is not chemical functionalization, it is still a widely used approach to improve functional properties of nanofibers. Therefore, we included this part in nanofiber functionalization.

  1. There is a clear discrepancy when discussing their own work vs the work of others - their own work is very detailed whereas other examples read more anecdotal.

Reply. We have now revised the discussion of other literature papers to provide a more insightful and balanced discussion.

  1. Define acronyms at first use such as EDC/NHS

Reply. We have revised the manuscript for acronyms and defined them all now. Thanks for spotting this mistake of ours.

  1. There are several run on sentences (4-5 lines) or a one sentence paragraphs.

Reply. We have now added more sentences as you suggested. Please check section 4.

  1. Should mention dual air jet spinning as a viable technique for vascular tissue engineering since it can simultaneously produce nano and micro fibers - section 3.4 example - see European Polymer Journal 96, 27-43

Reply. We added dual air jet spinning approach in 3.4 according to your suggestion. Please check Section 3.4.

Reviewer 2 Report

The authors reviewed an important field of tissue engineering discipline. The review is timely. However, the reviewer has the following concerns/comments:

  1. Polishing up the English is highly recommended;
  2. Figure 7A - change the oval shape to an egg shape;
  3. For a review article, the Future Outlook section is the most important. The current version should be re-written as it is confusing. As the lead researchers in the field, a clear direction should be given to guide the researchers in the field.

Author Response

Comments and Suggestions for Authors

The authors reviewed an important field of tissue engineering discipline. The review is timely.

However, the reviewer has the following concerns/comments:

1. Polishing up the English is highly recommended;

Reply. Thanks for the recommendation. We have now critically revised the paper. Of note, one of the co-authors is native speaker. The manuscript has been edited for readability throughout.

2. Figure 7A - change the oval shape to an egg shape;

Reply. We have changed to egg shape. Please check Figure 7A.

3. For a review article, the Future Outlook section is the most important. The current version should be re-written as it is confusing. As the lead researchers in the field, a clear direction should be given to guide the researchers in the field.

Reply. We have rewritten the future outlook section. We hope it is clearer now.

Reviewer 3 Report

The  manuscript review “Strategies to Improve Nanofibrous Scaffolds for Vascular Tissue
Engineering” by Tianyu Yao, Matthew B. Baker and Lorenzo Moroni addresses all the problematics in the field, is well written. So it is acceptable for publication once small editing mistakes are corrected:

  1. 13, line 522: electrospun
  2. 29, line 1342: Stanford 2017

Author Response

Comments and Suggestions for Authors

The  manuscript review “Strategies to Improve Nanofibrous Scaffolds for Vascular Tissue
Engineering” by Tianyu Yao, Matthew B. Baker and Lorenzo Moroni addresses all the problematics in the field, is well written. So it is acceptable for publication once small editing mistakes are corrected:

  1. 13, line 522: electrospun

Reply. We have corrected as requested.

  1. 29, line 1342: Stanford 2017

Reply. We have added the space. Please check in the reference.

Round 2

Reviewer 2 Report

None